# Intestinal Lipase Characterization in Common Snook (*Centropomus undecimalis*) Juveniles

**Bartolo Concha-Frías [1], Martha Gabriela Gaxiola-Cortes [2], Fanny Janet De la Cruz-Alvarado [3], Luis Daniel Jimenez Martinez [1], Emyr Saul Peña-Marin [3,4], Marcia Angélica Oliva-Arriagada [5], Joe Luis Arias-Moscoso [6,* and Carlos Alfonso Alvarez-González [3,***

[1] Laboratorio de Biología Molecular, DAMJ-UJAT, Carretera Estatal Libre Villahermosa-Comalcalco Km. 27+000 s/n Ranchería Ribera Alta, Jalpa de Méndez 86205, Mexico; bcfrias22@hotmail.com (B.C.-F.); luisd1984@hotmail.com (L.D.J.M.)

[2] Unidad Multidisciplinaria de Docencia e Investigación, Facultad de Ciencias, UNAM, Puerto de Abrigo s/n, Sisal 97356, Mexico; mggc@ciencias.unam.mx

[3] Laboratorio de Fisiología en Recursos Acuáticos, DACBIOL-Universidad Juárez Autónoma de Tabasco, Carretera Villahermosa Cárdenas Km 0.5, Villahermosa 86139, Mexico; fanylai@hotmail.com (F.J.D.l.C.-A.); ocemyr@yahoo.com.mx (E.S.P.-M.)

[4] Consejo Nacional de Ciencia y Tecnología, Av. Insurgentes Sur 1582, Col. Crédito Constructor, Del. Benito Juárez, Ciudad de México 03940, Mexico

[5] Laboratorio de Peces, Departamento de Acuicultura, Facultad de Ciencias del Mar, Universidad Católica de Chile, Campus Guayacán Larrondo 1281, Coquimbo 1780000, Chile; moliva@ucn.cl

[6] Department of Engineering, Technological National of Mexico, Technological Institute of the Yaqui Valley, Bacum 85276, Mexico

\* Correspondence: joearias@hotmail.com (J.L.A.-M.); alvarez_alfonso@hotmail.com (C.A.A.-G.); Tel.: +52-9933581500 (ext. 6480) (C.A.A.-G.)

**Abstract:** The common snook (*Centropomus undecimalis*) is a euryhaline fish with high commercial demand in the Mexican southeast, Caribbean, and South America. However, some aspects of its digestive physiology are still unknown, particularly in relation to lipid hydrolysis. Therefore, the characterization of the digestive lipase of this species was carried out. Our results show that the digestive lipase's optimal temperature is 35 °C, being stable between 25 and 35 °C, and shows maximum activity at pH 9, with stability between pH 5 and 8. Different degrees of inhibition were presented by Orlistat (61.4%), Ebelactone A (90.36%), Ebelactone B (75.9%), SDS 1% (80.7%), SDS 0.1% (73.5%), and SDS at 0.01% (34.9%). Orlistat and Ebelactone A and B completely inhibited the lipase band in the zymogram, but not SDS addition. Lipase showed a molecular weight of 43.8 kDa. The high lipase activities in the digestive tract indicate the importance of lipids in the diet of *C. undecimalis*.

**Keywords:** digestive lipase; inhibitors; temperature; pH; Centropomidae

## 1. Introduction

The common snook (*Centropomus undecimalis*, Bloch, 1792) is a carnivorous marine fish with high commercial importance in the Gulf of Mexico, part of the Caribbean, and South America [1,2], This situation has been accentuated by population growth and the projections made by the FAO for 2030 (FAO, 2020), and it is therefore necessary to integrate this species into aquaculture. However, studies on digestive physiology focusing on the functioning of the enzyme package in the digestive tract are required [3,4] to improve the assimilation of nutrients from the diet, with an increase in growth and lower costs [5–8]. Exogenous lipases from the microbiota of fish such as rainbow trout (*Oncorhynchus mykiss*) have been used and, in many cases, did not function as food additives [9], so the lack of knowledge about the digestive physiology of fish is a limitation for the formulation of aquatic food [10]. In *C. undecimalis*, some studies have been carried out on the importance

of digestive enzymes such as the activity and early expression of proteases, lipases, and α-amylase [11,12]. However, lipases require special attention since they are essential enzymes of the digestive system, where catalytic activity can be evaluated by fast, reliable, specific, selective, and sensitive analytical methods [13].

Typically, carnivorous fish feed prey with high lipid content; consequently, it is considered that they require more significant lipase activity compared to other omnivorous species [7,14]. According to Refs. [3,8], lipids are essential in the structure of cell membranes by maintaining flexibility and permeability, in the storage of energy and fatty acids, and in participating in cell signaling [15,16].

Carboxyl ester lipase (CEL) are a group of water-soluble carboxylic ester hydrolase enzymes that move between the cells of an organism [17] and carry out the hydrolysis of triglycerides to monoglycerides to release fatty acids [18,19] by breaking lipid bonds, acting between the aqueous and organic phases [13,20,21]. They are classified into two groups: (1) Esterases (EC 3.1.1.1), which perform the excision of low-molecular-weight fatty acids and have solubilizing capacity, acting on simple ester bonds by catalyzing the rupture of ester bonds of vitamins, phospholipids, triglycerides (TGS), and cholesterol esters. They are active in water-insoluble lipid substrates; (2) True (EC 3.1.1.3 acyl triglyceride hydrolase) or bile-salt-dependent lipases, the dominant lipase in fish [22,23], which are synthesized in the pancreas, requiring bile salts for their proper functioning in concentrations between 25 mM and 250 mM, such as sodium taurocholate ($C_{26}H_{44}NNaO_7S$), sodium taurodeoxycholate ($C_{26}H_{44}NNaO_6S$), and cholic acid ($C_{24}H_{40}O_5$), produced and/or recycled in the liver in 60 to 95% as a product of immunoglobulin degradation, lipids such as cholesterol and steroids, which are essential for the solubilization, hydrolysis, and adsorption of lipids through enterocytes and preserving their denaturation and considerably increasing their activity at pH 8 [24–27]. Lipase is a unique polypeptide chain that folds into a large N-terminal domain belonging to the fold α/β-hydrolase and a smaller C-terminal domain containing a catalytic triad of serine, aspartic acid, and histidine that is analogous to serine, thus favoring the adsorption of the substrate on the intestinal walls of fish, allowing them to act on poorly soluble substrates [28]. They are currently used in the food, pharmaceutical, detergent, biotensive, and optically active compound industries. For these reasons, fish is being studied as a fundamental source of lipase production to improve industrial processes [29,30].

Lipase activity and its characterization have been reported in several fish species such as Yellowfin tuna (*Thunnus albacares*), Longtail tuna (*Thunnus tonggol*), Skipjack tuna (*Katsuwonus pelamis*) [4], Siamese fighting fish (*Betta splendens*) [14], European perch (*Perca fluviatilis*) and Arctic char (*Salvelinus alpinus*) [7], and Mozambique tilapia (*Oreochromis mossambicus*) [8]. However, in *C. undecimalis*, this type of study has not been carried out; consequently, our main objective is to characterize bile-salt-activated lipase by biochemical and electrophoretic techniques in the intestine of *C. undecimalis*, and it is here that they exert their catalytic activity.

## 2. Material and Methods

### 2.1. Ethical Statement

Fish were handled in compliance with the standards for the good welfare practices of laboratory animals from the Norma Mexicana NOM-062-ZOO-1999 de la Secretaría de Agricultura, Gandara, Desarrollo Rural, Pesca y Alimentación.

### 2.2. Capture and Maintenance of Juveniles

For this study, a total of 50 wild juveniles (3.22 ± 0.16 g and 7.25 ± 0.14 cm) were captured in the month of July from the natural environment, with conical mosquito nets 15 m long × 3 m high, in Arroyo Verde, Comalcalco, Tabasco Mexico. The fish were transferred in containers with constant aeration to the Laboratorio de Fisiología en Recursos Acuáticos by the División Académica de Ciencias Biológicas of the Universidad Juárez Autónoma de Tabasco, México, and fed with tilapia (*Oreochromis niloticus*) fingerlings, in

circular plastic ponds at 28 °C, for 7 days. Before slaughter by overdose of MS-222 (tricaine methanesulfonate), they were starved for 48 h.

### 2.3. Enzyme Extract and Enzymatic Technique

All the fish were sacrificed, and their intestines were dissected under ice conditions (4 °C). The organs were weighed, then the organ set was homogenized in 50 mM Tris-HCl + 25 mM CaCl$_2$, pH 7.5 buffer, in a ratio of 1:5, and centrifuged at 16,000× *g* for 30 min at 4 °C. Subsequently, the supernatant was removed and stored at −20 °C for subsequent analysis.

Activity was measured using the method described by Versaw et al. [31], where the reaction mix consisted of 20 μL of enzyme extract in 200 μL of 100 mM sodium taurocholate and 1.9 mL of 50 mM Tris-HCl buffer at pH 7.5. The mix was incubated at room temperature for 5 min, and the reaction was started with 20 μL of β-naphthyl caprylate (20 mM) for 30 min at 35 °C. Then, 20 μL of fast blue was added at 100 mM and incubated for 5 min at room temperature. The reaction was stopped with 200 μL of trichloroacetic acid (TCA; 0.72 N) and clarified with 2.71 mL of ethanol ethyl acetate at 1:1 *v/v*. Finally, the absorbance was read at 540 nm in quartz cuvettes. The concentration of soluble protein was determined using the Bradford [32] technique with bovine serum albumin as the standard. Calculation of specific activity of individual extracts was determined using the following equations: (1) units mL$^{-1}$ = [Δabs × final reaction volume (mL)] × [MEC × time (min) × extract volume (mL)] and (2) mg protein units$^{-1}$ = [units per mL] × [mg of soluble protein]$^{-1}$, where Δ abs = Increase in absorbance at a given wavelength; Final volume = Final volume of the reaction (mL); MEC = Molar extinction coefficient (MEC) calculated from the regression line of 2-naphthol (0.02 mL mg$^{-1}$ cm$^{-1}$); Time = incubation time of the catalyzed reaction (min); Extra-spectrum volume = Volume of multienzyme extract (mL). All tests were performed in triplicate.

### 2.4. Temperature and pH Effect on Enzymatic Activity

The effect of temperature on lipase activity was determined with universal buffer [33] at pH 9 in the range between 25 and 65 °C and using the technique previously described. The effect of pH on lipase activity was determined with universal buffer [33], varying the pH between 2 and 12. All assays were performed in triplicate. The residual activity was determined at different times (30, 60, 90, and 120 min of pre-incubation), varying temperatures (25, 35, 35, 55 °C) and pH (2, 3, 4, 5, 6, 7, 8, 9, 10 and 11) compared to a control without pre-incubation using the technique of Versaw et al. [31]. All tests were performed in triplicate.

### 2.5. Inhibitor Effect on Enzymatic Activity

The effect of inhibitors on lipase activity was determined by a multi-enzymatic extract pre-incubated with Orlistat 2.6 mM, microbial Ebelactone A 1 mM, and microbial Ebelactone B 1 mM [34] and in increasing amounts (0.01, 0.1, and 1%) of sodium lauryl doudecylsulfate (SDS) using the technique proposed by Görgün and Akpınar [15], compared to a control without pre-incubation, using the same technique as Versaw et al. [31], as previously described. All tests were performed in triplicate.

### 2.6. Zymogram Analysis

Electrophoresis was performed under native conditions following the technique proposed by Davis [35] on 10% polyacrylamide gels in buffer Tris-HCl 1.5 mol L$^{-1}$ at pH 8.8. The electrophoretic was run at 80 V for 15 min and then increased to 120 V for 2 h at 4 °C. As substrate was used a solution of β-naphthyl caprylate (200 mmol L$^{-1}$), where gel was washed for 30 min. At the end, fast blue solution (100 mmol L$^{-1}$) was added, and it was incubated at 25 °C until the lipase activity bands were observed. The same four inhibitors previously mentioned were used, which were pre-incubated in a 1:1 ratio (enzyme/inhibitor) for 1 h. Molecular weight markers Bio Basic Inc. (Markham, ON, Canada) BM523 and Quantity One 1-D Analysis Software from Bio-Rad (Hercules, CA, USA) (phos-

phorylase 97 kDa, bovine serum albumin 66 kDa, Ovalbumin 45 kDa, carbonic anhydrase 29 kDa, trypsinogen 24 kDa, and SBTI 20 kDa) were used to calculate the molecular weight of the band with activity.

### 2.7. Statistical Analysis

Normality and homoscedasticity were corroborated for the values of enzymatic activity > One-way analysis of variance was applied, followed by Duncan's tests to determine the differences between the treatments. For all statistical tests, a significance value of $p < 0.05$ was used. All statistical analyzes were performed using STATISTICA$^{TM}$ v. 7.0 software (Statsoft, Tulsa, OK, USA). In addition, Sigma Plot 12.0 software was used to draw the graphics.

### 3. Results

### 3.1. Temperature and pH Optimum and Stability

The specific activity is expressed in units per milligram of protein and represents the amount of enzyme that catalyzes the formation of one µmol of product per minute under conditions of substrate saturation. In this study, the optimum temperature of lipases was detected at 35 °C (0.27 ± 0.04) (Figure 1) with a drop-in activity from 45 °C (0.218 ± 0.01), showing statistically significant differences in all treatments for $p < 0.05$. In thermal stability, the highest activity was observed at 25 °C at 60 min of incubation (154.24 ± 1.71), followed by high activity at 35 °C (116.16 ± 3.69) above 100% of the residual activity, showing a decrease in activity at 90 min of incubation at 25 °C (116.8 ± 1.21) and at 35 °C (93.76 ± 2.64), respectively, showing practically no activity at temperatures higher than 35 °C (Table 1).

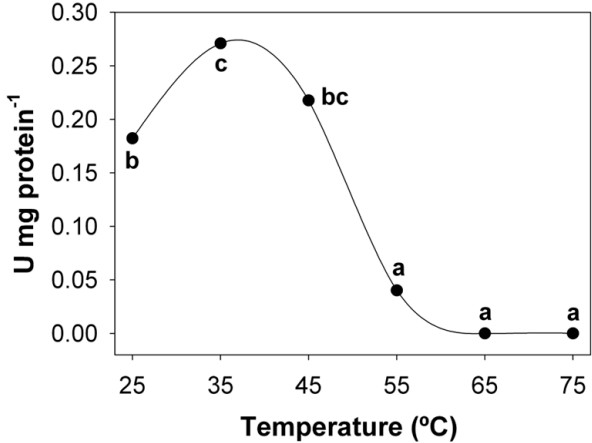

**Figure 1.** Optimum temperature of intestinal lipases in the common snook *C. undecimalis*. Different letters indicate significant differences between temperatures.

**Table 1.** Stability at different temperatures of intestinal lipases in the common snook *C. undecimalis*.

| °C | 0 min | 30 min | 60 min | 90 min |
|---|---|---|---|---|
| 25 | 100 ± 0.00 [c] | 68.16 ± 1.02 [d] | 154.24 ± 1.71 [a] | 116.80 ± 1.21 [b] |
| 35 | 100 ± 0.00 [c] | 58.24 ± 3.01 [d] | 116.16 ± 3.69 [a] | 93.76 ± 2.64 [b] |
| 45 | 100 ± 0.00 [a] | 0.00 ± 0.00 [c] | 6.56 ± 0.09 [b] | 4.16 ± 0.05 [b] |
| 55 | 100 ± 0.00 [a] | 0.00 ± 0.00 [b] | 0.00 ± 0.00 [b] | 0.00 ± 0.00 [b] |

Means in the same row with different superscripts are significantly different ($p < 0.05$).

The highest activity was presented at pH 8 (1.96 ± 0.08) and pH 9 (1.98 ± 0.08), respectively, with no statistically significant differences between these two treatments, with low activities at pH 7 (1.31 ± 0.07) and pH 6 (0.96 ± 0.08), showing statistically significant differences between them and those mentioned above (Figure 2). Stability was shown at pH 7 (157.16 ± 12.59) and pH 8 (156.92 ± 8.87) at 60 min of incubation, even exceeding the

residual activity without showing statistically significant differences for these two points, with a drop at 90 min for pH 7 (108.95 ± 3.76) and pH 8 (112.05 ± 1.29), respectively, while at pH 5 and pH 6, the activity decreased after 30 min of incubation. The residual activity of all the other pHs analyzed showed activities below 100% of the residual activity (Table 2).

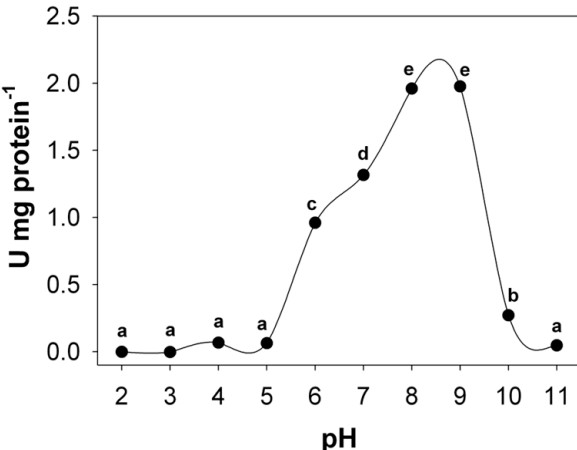

**Figure 2.** Optimum pH of intestinal lipases in the common snook *C. undecimalis*. Different letters indicate significant differences between pH values.

**Table 2.** Stability at different pH of intestinal lipases in the common snook *C. undecimalis*.

| pH | 0 min | 30 min | 60 min | 90 min |
|----|-------|--------|--------|--------|
| 2 | 100 ± 0.00 [a] | 0.00 ± 0.04 [d] | 46.18 ± 3.01 [b] | 11.34 ± 0.22 [c] |
| 3 | 100 ± 0.00 [a] | 12.29 ± 0.82 [c] | 23.87 ± 1.11 [b] | 10.74 ± 0.50 [c] |
| 4 | 100 ± 0.00 [a] | 80.55 ± 1.04 [b] | 45.35 ± 1.02 [c] | 95.59 ± 1.21 [a] |
| 5 | 100 ± 0.00 [c] | 122.32 ± 3.04 [b] | 101.31 ± 1.03 [c] | 139.98 ± 2.02 [a] |
| 6 | 100 ± 0.00 [d] | 139.14 ± 5.01 [a] | 107.40 ± 4.06 [c] | 118.74 ± 4.01 [b] |
| 7 | 100 ± 0.00 [c] | 144.87 ± 4.02 [a] | 157.16 ± 12.59 [a] | 108.95 ± 3.76 [b] |
| 8 | 100 ± 0.00 [d] | 137.95 ± 6.01 [b] | 156.92 ± 8.87 [a] | 112.05 ± 1.29 [c] |
| 9 | 100 ± 0.00 [a] | 70.29 ± 2.01 [c] | 76.37 ± 1.04 [b] | 53.34 ± 0.72 [d] |
| 10 | 100 ± 0.00 [a] | 101.91 ± 1.02 [a] | 58.23 ± 1.00 [b] | 30.31 ± 0.13 [c] |
| 11 | 100 ± 0.00 [a] | 36.64 ± 1.03 [d] | 66.71 ± 1.22 [b] | 57.04 ± 2.23 [c] |

Means in the same row with different superscripts are significantly different ($p < 0.05$).

### 3.2. Inhibitor Effect on Lipase Activity

The presence of inhibitors shows different degrees of inhibition in the activity of lipase, where the highest degree of inhibition was observed when the enzyme was exposed to the presence of Ebelactone A (89.8 ± 8.6%), followed by SDS at 1% (80.2 ± 12.7%) and Ebelactone B (75.7 ± 4.7%), showing significant differences between Ebelactone B and SDS at 1%. Likewise, SDS at 0.1% (73.2 ± 6.9%) showed a greater effect on relative activity than SDS at 0.01% (35.3 ± 1.8%), while Orlistat showed the least inhibitory effect (61.6 ± 10.1%) in relation to the group of inhibitors used in this study, showing statistically significant differences between treatments ($p < 0.05$) (Figure 3).

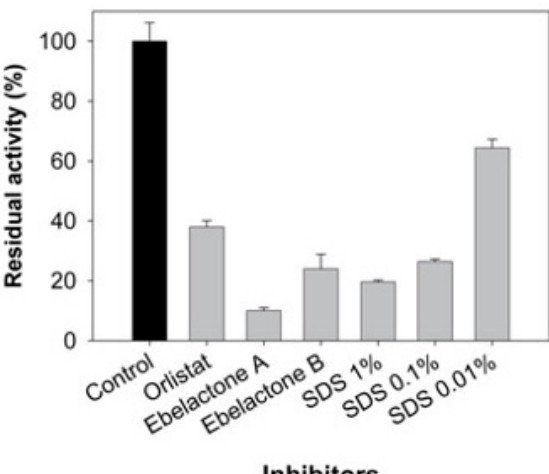

**Figure 3.** Effect of inhibitors on lipolytic activity in juveniles of common snook *C. undecimalis*.

*3.3. Zymogram Analysis*

Zymogram of lipase shows a single band with a molecular weight of 43.8 kDa, that was not affected by different concentrations of SDS, while Orlistat, Ebelactone A, and Ebelactone B completely inhibited bands (Figure 4).

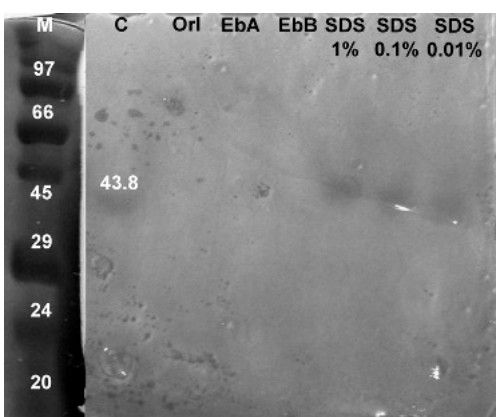

**Figure 4.** Zymogram of inhibition under native conditions on 10% polyacrylamide gels of lipolytic activity in juveniles of common snook *C. undecimalis*. M (molecular weight marker: phosphorylase 97 kDa, bovine serum albumin 66 Kda, Ovalbumin 45 Kda, carbonic anhydrase 29 Kda, trypsinogen 24 Kda, and SBTI 20 kDa). Control (lipases without inhibitor). Inhibitors: Orl (Orlistat), EbA (Ebelactone A), EbE (Ebelactone B), SDS (sodium dodecyl sulfate).

## 4. Discussion

Lipases are responsible for carrying out lipolysis [14]. Lipids contained in the diet are their primary source of energy and play a vital role in the composition of the cell membrane and signaling, where fish consume lipid-rich foods [3]. In this study, *C. undecimalis* showed higher lipase activity at 35 °C, like that reported for hoki (*Macruronus novaezelandiae*) (35 °C) [23], bluefin tuna (*Thunnus orientalis*) (40 °C), totoaba (*Totoaba macdonaldi*) (45 °C), striped bass (*Morone saxatilis*) (35 °C) [25], and Atlantic cod (*Gadus morhua*) (25 to 30 °C) [30]. Ref. [24] reports optimal temperature in Catla (*Catla catla*) lipases at 20 °C, although species such as Chinook salmon (*Oncorhynchus tshawytscha*) present two optimal temperatures (35–40 °C and 50 °C) that possibly indicate the presence of lipases isoforms [23]. Ref. [22] indicates that species such as *O. niloticus*, *G. morhua*, Indian oil sardine (*Sardinella longiceps*), and picked dogfish (*Squalus acanthias*) show optimal activity between 25 °C and 37 °C, except for flathead grey mullet (*Mugil cephalus*), which shows more significant activity at 50 °C, as well as gilthead seabream (*Sparus aurata*) [36]. Thus, the optimal activity of lipases

varies with the temperature of the fish habitat [37], culture parameters, and carbon and nitrogen sources [29].

*C. undecimalis* requires an optimal culture temperature between 25 and 29 °C, with mortalities at temperatures of 10 °C and 35 °C, respectively [2]. We find that the temperature stability was between 25 °C and 35 °C for *C. undecimalis*, losing significant activity when incubated at temperatures above 35 °C, which is related to the mortalities. These stability data are like those reported for *O. tshawytscha* and *M. novaezelandiae* [23] and *C. catla* (20 °C) [24]. In contrast, Kurtovic et al. [22] reports stability fluctuations between 30 °C and 50 °C in species such as *O. niloticus*, *G. morhua*, *S. longiceps*, *S. acanthius*, and *M. cephalus*. Nolasco et al. [32] reports optimal activities at 40 °C in *S. aurata*, such as the results obtained by González-Felix et al. [38] in *T. macdonaldi* at 45 °C.

Currently, there are no reported data on the gastrointestinal pH of *C. undecimalis*; however, most of the fish studied show an intestinal pH between 6.7 and 8.5 [39–44]. This parameter may be related to the catalytic activity of lipases. Regarding pH activity, *C. undecimalis* showed the highest intestinal lipase activity at pH 8 and 9. Similar data were reported for *S. aurata* [36]; *T. orientalis*, *T. macdonaldi*, and *M. saxatilis* [25]; juveniles of *O. niloticus* [16]; and *O. tshawytscha*, *M. novaezelandiae* and *T. macdonaldi* [23,29]; however, activities at different pH have also been reported such as in *C. catla* at 7.8 [24]. Thongprajukaew et al. [14] report three different peaks at 7, 8, and 11, according to the development stage of *B. splendens*. Prasertsan et al. [4] report maximum activities of digestive lipases at pH 10 in *T. albacares* and *K. pelamis*, and at pH 9 in *T. tonggol*, indicating that the optimum pH of lipase can vary in species between 6.5 and 9 [22]. pH stability was observed in the range of 5 to 8, with greater stability at pH 7 and 8, which was reported for *O. tshawytscha*, *M. novaezelandiae* [23], and *S. aurata* [36]. Similarly, in *S. longiceps*, stability was reported between pH 5 and 9.5, while *O. niloticus* showed stability between 6.5 and 8.5 [22]. Areekijseree et al. [45] indicates that developmental stage and environment can modify optimal activities at different temperatures and pH, as observed in *Hyriopsis bialatus*. Knowing the thermal and pH stability in fish is vital, since these data reflect the optimal digestive activity that these organisms possess over a long period of time under common environmental conditions in their environment. In contrast, optimal temperature and pH indicate maximum proteolytic activity before denaturation of enzymes, which is achieved in a short time.

Orlistat is a tetrahydrolipstatin (THL) that belongs to the β-lactone group derived by the hydrogenation of lipstatin produced by the fungus *Streptomyces toxytricini*. It is known as a potent, irreversible, and specific inhibitor of gastric and pancreatic lipases [46], where the action mode is through the formation of a double bond with the serine catalytic residue of lipase in the β-lactone group, blocking the active site and preventing the hydrolysis of triglycerides from freeing fatty acids, which in the physiological process are eliminated through the faces [36,47]. This chemical compound has been used mainly in humans for treatments of obesity and pancreatitis [48,49]; however, it reduces the incorporation of n-3 long-chain polyunsaturated fatty acids into the blood and tissues of rats [50]. Orlistat in this study inhibited lipase activity by 61.6%, and its inhibitory action was also verified in fish, while in *Centropomus viridis*, it inhibited 71.16% at a concentration of 1 mM. This high inhibition at a lower concentration may be since [51] worked with larvae, and at that stage, they found great activity of neutral lipases, while in juveniles, they were already absent. To this, it can also be added that there are variations in the activities of lipases, even though they are related species, even in the same species with a different diet, because of genetic change [52].

Ebelactone A and B are a small group of β-lactone esterase-inhibiting enzymes, lipases, and N-formylmethionine aminopeptidases located on the cell membrane and synthesized by the fungus *S. aburaviensis*. In this sense, the inhibitory power of Ebelactone B shows a decrease in absorption of lipids by the intestinal wall from the diet in mice [53,54], where the β-lactone group inactivates the active lipase site, as previously described for Orlistat [49]. Ebelactone A inhibited lipase activity by 89.8% in our trial, while Ebelactone B showed less

inhibitory power at 75.7% [55]. On the other hand, cationic surfactants such as quaternary ammonium salts and anionics such as sodium dodecyl sulfate (SDS) are components with the highest inhibition in lipase activity, such as diethyl p-nitrophenyl phosphate, which affects the active site of serine through irreversible inhibition [22]. SDS at 1% inactive 80.2% of the activity in this study, SDS at 0.1% inactive 73.2% of lipase activity, and SDS at 0.01% inactive 35.3% of lipase activity, showing a dose-dependent relationship. Görgün and Akpınar [15] reports a total loss of lipase activity in *Cyprinus carpio* L. using 0.5% SDS.

The molecular weight of the digestive lipase found in *C. undecimalis* was 43.8 kDa. Fish studies show digestive lipases with molecular weights between 46 and 64 kDa [22]. These studies include those published for *O. niloticus* (46 kDa) [37], bighead carp (*Aristichthys nobilis*) (127.9 kDa), hybrid sturgeon (*Huso dauricus* ♀X *Acipenser schrenki* Brandt ♂) (40.5 kDa) [3], *C. catla* (70 kDa) [24], Round sardinella (*Sardinella aurita*) (43 kDa) [56], and *T. macdonaldi* (70 kDa) [38]. Villanueva-Gutiérrez et al. [29] report two molecular weights for *T. macdonaldi* lipase (70.2 kDa and 47.5 kDa), suggesting the presence of two different forms: (a) the uncleaved form and (b) the final form of a pancreatic lipase dependent on colipase since the activity was detected without the presence of bile salts complementing the inhibited activity. Other studies in *O. tshawytscha* report molecular weights of 79.6 and 54.9 kDa, and *M. novaezelandiae* shows 44.6 kDa [23]. Kurtovic et al. [22] indicate that a molecular weight of 57 kDa may suggest the presence of a carboxyl ester lipase, supported by results shown by [57] for *S. longiceps* (54–57 kDa). Therefore, molecular weights depend on the species, stages of development, eating habits, and tissues.

Although most of the studies carried out with lipase activities indicate that herbivore fish show higher activity than carnivorous, similar levels of activity among them have been reported without finding significant differences [58]. Other studies have indicated that sea bass in the marine environment obtain their energy from lipids, proteins, and carbohydrates, indicating that it would be possible to find a higher activity of lipases in this environment than in freshwater fish that use proteins as energy sources. However, [59] found no differences when studying lipase activities in carnivorous and herbivorous fish in two different environments, although this differs from that detected by [60], who reported higher lipase activity in the herbivorous species monkeyface prickleback (*Cebidichthys violaceus*) and rock prickleback (*Xiphister mucosus*) than in the carnivores black prickleback (*Xiphister atropurpureus*) and high cockscomb (*Anoplarchus purpurescens*). Horn et al. [41] reports that the highest activity of lipases takes place in the anterior intestine, although Hariati et al. [61] indicate that high activity can occur in the anterior or posterior intestine depending on the time of year and, therefore, on the diet, which would increase activity along with growth, as observed in gachua (*Channa gachua*). Thus, the environment and bioavailability of nutrients in the fish diet will regulate lipase activity [10,26,62–64], and studies with pH STAT could help to identify the catalytic differences of lipases for a substrate of plant or animal origin.

## 5. Conclusions

The characterization of the lipase of common snook (*Centropomus undecimalis*) is vital to determine the enzymatic activity and its relationship with pH and temperature. This knowledge can help us in future studies to determine the nutritional balance of diets applied to the cultivation of this species. In our study, the lipase enzyme has a molecular weight of 43.8 kDa, with an optimal temperature of 35 °C and an optimal pH of 9, presenting different sensitivities to the inhibitors. In this study, Ebelactone A is the substance that inhibited the most lipase activity. Further studies are needed to determine suitable lipid sources that will be hydrolyzed by *C. undecimalis* lipases and contribute to the proper nutrition of this species.

**Author Contributions:** Conceptualization, B.C.-F., M.G.G.-C. and F.J.D.l.C.-A.; methodology, J.L.A.-M., L.D.J.M. and E.S.P.-M.; software, E.S.P.-M. and M.A.O.-A.; formal analysis, B.C.-F. and F.J.D.l.C.-A.; investigation, C.A.A.-G., M.A.O.-A. and L.D.J.M.; writing—original draft preparation, B.C.-F.; writing— review and editing, B.C.-F., J.L.A.-M. and C.A.A.-G.; project administration and supervision, C.A.A.-G. All authors have read and agreed to the published version of the manuscript.

**Funding:** This research received no external funding.

**Institutional Review Board Statement:** Animals were handled in compliance with the Norma Official Mexicana NOM-062-ZOO-1999 from Secretaría de Agricultura, Ganadería, Desarrollo Rural, Pesca y Alimentación, Mexican standards for good welfare practices of laboratory animals.

**Data Availability Statement:** Data are available from the authors upon request.

**Acknowledgments:** We thank the editors and the reviewers of this paper for their constructive feedback.

**Conflicts of Interest:** The authors declare that there are no conflicts of interest regarding the publication of this article.

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
