# Peer review of "Intestinal Lipase Characterization in Common Snook (Centropomus undecimalis) Juveniles"

_fishes, doi:10.3390/fishes7030107_

Round 1

Reviewer 1 Report

The authors have worked to increase the quality of the manuscript. However there are still some details that should be changed:

  • in the abstract, the sentence "However, some aspects of its digestive physiology are still unknown, allowing us to develop a commercial diet" seems to be wrong. Not knowing the digestive physiology of the species cannot promote the development of a diet.
  • some aspects of methodology are missing: 1) in the zymogram at what temperature was the gel incubated to reveal the lipase activity? 2) in the inhibitors test, what was the volume of enzyne extract and inhibitor used? 3) Rearing temperature of the fish? 4) why the authors refer to a "multi enzymatic extract" in the inhibitors' effects section?

English should be improved

Author Response

In the abstract, the sentence "However, some aspects of its digestive physiology are still unknown, allowing us to develop a commercial diet" seems to be wrong. Not knowing the digestive physiology of the species cannot promote the development of a diet.

Response: Lines 20 and 21 were modified

Some aspects of the methodology are missing:

1) in the zymogram at what temperature was the gel incubated to reveal the lipase activity?

Response: Line 323 was modified

2) in the inhibitors test, what was the volume of enzyme extract and inhibitor used?

Response: The volume of enzyme extract used was 20 microliters, while 1 microliter of inhibitor was used per sample

3) Rearing temperature of the fish?

Response: Line 271 was modified

4) Why the authors refer to a "multi enzymatic extract" in the inhibitors' effects section?

Response: The term "multienzyme extract" is due to the fact that in the extraction sample there is a concentrate of enzymes of different types (Lipases, proteases, phosphatases, etc), and the inhibitors used are specific for lipases, which allows us to identify their presence and degree of participation during the catabolism of nutrients, a technique that allows us to exclusively isolate lipases was not applied.

English should be improved.

Response: The English language was checked according to your recommendation. Thanks.

Reviewer 2 Report

Review for the paper "Intestinal lipase characterization in common snook (Centropomus undecimalis) juveniles" by Bartolo Concha-Frías, Martha Gabriela Gaxiola Cortes, Fanny Janet De la Cruz-Alvarado, Luis Daniel Jimenez-Martinez, Emyr Saul Peña-Marin, Marcia Angélica Oliva-Arriagada, Joe Luis Arias-Moscoso and Carlos Alfonso Alvarez-González submitted to "Fishes".

General comment.

Several recent studies have covered various aspects of fish development, with emphasis on feed acquisition and digestive processes illustrating the advances made in knowledge from different perspectives, highlighting the essential role of feeding and digestive organs, as well as behavior and the digestive machinery that enables rapid growth, in addition to the anatomical status and transformations exhibited by fish during their growth. It is obvious that better knowledge of fish feeding behavior and digestive physiology will provide a basis for the optimization of diets and feeding protocols and will eventually improve growth rates and survival in many common aquaculture species. An integrated understanding of the various factors and events interacting in food acquisition and digestion is necessary for the design of diets that meet the requirements for optimal ingestion, digestion and absorption in fish. Although there is substantial progress regarding the digestive physiology of various fish there is a lack of knowledge concerning the species which have became important for aquaculture in recent years. One of such species is the common snook Centropomus undecimalis distributed in the Gulf of Mexico, Caribbean, and South America. The authors studied the digestive lipase of this species to obtain new data which may be useful for the development of a commercial diet for Centropomus undecimalis. The authors found the optimal range for temperature and pH and tested different substances for their inhibition effects for the digestive lipase. The authors suggested the importance of lipids in the diet of common snook.

The present study contributes to our knowledge on the biology of Centropomus undecimalis and may be of interest to ichthyologists. Standard analytical methods were used in the study. The data were compared using ANOVA. Main results are illustrated with relevant Figures and Tables. The discussion is focused on the main findings.

Recommendation

Figures 1, 2. In the captions, the authors should specify what the letters show.

Specific remarks.

L 76. Consider replacing “Lipase activity and its characterization has been reported” to “Lipase activity and its characterization have been reported”

L 140–149, 155–159 and throughout the text. The authors should use italics when referring Latin names of species.

L 148. Consider replacing “as well than gilthead” to “as well as gilthead”

L 155. Consider replacing “These stability data are like those reported” to “These stability data are similar to those reported”

L 162. Consider deleting “(Horn et al., 2006)”

L 209. Consider replacing “finding  a  relationship  dose-response” to “showing  a  dose-dependent relationship  ”

L 218. Consider replacing “suggesting  the  presence  two” to “suggesting  the  presence of  two”

L 224. Consider replacing “weights depending on” to “weights depend on”

L 230. Consider replacing “fish the energy substrate is proteins” to “fish which use proteins as energy sources”

L 277. Consider deleting “(Stauffer, 1989)”

L 282. Consider replacing “using Versaw et al. (1989) [63]” to “using Versaw et al. [66]”

L 289. Consider replacing “using same Versaw et al. (1989) technique previous described” to “using same the Versaw et al. technique previously described”

L 306. Consider replacing “analysis of variance of one way and later” to “one-way analysis of variance followed by”

Author Response

General comment.

Several recent studies have covered various aspects of fish development, with emphasis on feed acquisition and digestive processes illustrating the advances made in knowledge from different perspectives, highlighting the essential role of feeding and digestive organs, as well as behavior and the digestive machinery that enables rapid growth, in addition to the anatomical status and transformations exhibited by fish during their growth. It is obvious that better knowledge of fish feeding behavior and digestive physiology will provide a basis for the optimization of diets and feeding protocols and will eventually improve growth rates and survival in many common aquaculture species. An integrated understanding of the various factors and events interacting in food acquisition and digestion is necessary for the design of diets that meet the requirements for optimal ingestion, digestion and absorption in fish. Although there is substantial progress regarding the digestive physiology of various fish there is a lack of knowledge concerning the species which have become important for aquaculture in recent years. One of such species is the common snook Centropomus undecimalis distributed in the Gulf of Mexico, Caribbean, and South America. The authors studied the digestive lipase of this species to obtain new data which may be useful for the development of a commercial diet for Centropomus undecimalis. The authors found the optimal range for temperature and pH and tested different substances for their inhibition effects for the digestive lipase. The authors suggested the importance of lipids in the diet of common snook.

The present study contributes to our knowledge on the biology of Centropomus undecimalis and may be of interest to ichthyologists. Standard analytical methods were used in the study. The data were compared using ANOVA. Main results are illustrated with relevant Figures and Tables. The discussion is focused on the main findings.

Response: Thanks for your comments.

Recommendation

Figures 1, 2. In the captions, the authors should specify what the letters show.

Response: Lines 98, 116 and 117 were modified

Specific remarks.

L 76. Consider replacing “Lipase activity and its characterization has been reported” to “Lipase activity and its characterization have been reported”

Response: Line 76 was modified

L 140–149, 155–159 and throughout the text. The authors should use italics when referring Latin names of species.

Response: The indicated observations were made

L 148. Consider replacing “as well than gilthead” to “as well as gilthead”

Response: Line 163 was modified

L 155. Consider replacing “These stability data are like those reported” to “These stability data are similar to those reported”

Response: Line 170 was modified

L 162. Consider deleting “(Horn et al., 2006)”

Response: The phrase “(Horn et al., 2006)” was deleted

L 209. Consider replacing “finding a relationship dose-response” to “showing a dose-dependent relationship”

L 218. Consider replacing “suggesting the presence two” to “suggesting the presence of two”

Response: Lines 224 and 232 were modified

L 224. Consider replacing “weights depending on” to “weights depend on”

Response: Line 238 was modified

L 230. Consider replacing “fish the energy substrate is proteins” to “fish which use proteins as energy sources”

Response: Line 245 was modified

L 277. Consider deleting “(Stauffer, 1989)”

Response: The phrase “(Stauffer, 1989)” was deleted

L 282. Consider replacing “using Versaw et al. (1989) [63]” to “using Versaw et al. [66]”

Response: Line 306 was modified

L 289. Consider replacing “using same Versaw et al. (1989) technique previous described” to “using same the Versaw et al. technique previously described”

Response: Line 314 was modified

L 306. Consider replacing “analysis of variance of one way and later” to “one-way analysis of variance followed by”

Response: Line 334 was modified

Reviewer 3 Report

Paper is well conceptualized and written.

Only minor corrections are necessary, especialy species names that should be in italic (almost evrywhere - only in abstract and introdiction you have italic).

Lines 227-230. Correct the sentence (the last part doesn't make a sence; possibly where  the energy substrate are proteins).

Lines  239-242- "Thus",...., and then "therefore". Could be two sentences.

Author Response

Reviewer 3

Only minor corrections are necessary, especially species names that should be in italic (almost everywhere - only in abstract and introduction you have italic).

Response: The indicated observations were made

Lines 227-230. Correct the sentence (the last part doesn't make a since; possibly where the energy substrate are proteins).

Response: Line 245 was modified

Lines 239-242- "Thus",...., and then "therefore". Could be two sentences.

Response: Line 255 was modified

This manuscript is a resubmission of an earlier submission. The following is a list of the peer review reports and author responses from that submission.

Round 1

Reviewer 1 Report

The present work presents the characterization of digestive lipase in the common snook (Centropomus undecimalis). The objective of the study is important to correctly design feeds for the species, however there are some points to be addressed:

  • Why optimum temperature was only addressed between 25 and 75 ºC? Usually, temperature curves start with lower temperatures to ensure that there are not another peak.
  • Why fish were starved before sampling? Since pancreatic enzymes secretion, such as lipase secretion, is stimulated by feed ingestion, enzymes activity levels after 48h starving are expected to be at their minimum. 
  • Why was chosen these inhibitors' concentrations? Did the authors discard that higher concentration may induce higher inhibition?
  • Along the results section, authors refer several times to statistical differences, however, statistical data are not included in the paper. The reviewer suggests including in the figures statistical differences between temperatures, pHs, or inhibitors. 
  • In the thermal stability figure, there is a decrease in activity after 30min at 25 and 35ºC higher than the decrease observed after 90min. Is it statistically significant?
  • The description of pH stability results is so confusing. The reviewer suggests rewriting it.
  • How do authors explain that inhibitors' effects in the zymogram are not in line with the inhibition analyses? In "Inhibitor effect on lipase activity" section it is described that SDS 1% inhibited more lipase activity than Ebelactone B and Orlistat, however in the zymogram, it was observed a band with SDS 1% and not with EbB or Orl, suggesting higher inhibition capacity. 
  • The reviewer that discussion section needs to be improved. The authors cite several previous studies, but there is a lack of discussion itself, following I present some examples: 
  • L153, authors mention that optimal Tº varies with fish habitat, however do not mention any information about common snook habitat or the range of temperature in which fish are found, with might be closely related with the physiological adaptation.
  • About pH stability, do authors know the pH values of the intestine of the common snook? It may be relevant to discuss the optimum pH curve. 
  • Scarce discussion about lipase inhibition results in other fish species.
  • L205 reported total lose of lipase activity, please specify in which species.
  • Regarding the following conclusion: "The high digestive lipase activities of C. undecimalis indicate the importance of lipids in this species' diet" The authors do not compare or discuss activity levels with other studies on related species to state that activity levels are high.

Citation format and overall English should be reviewed.

Reviewer 2 Report

The manuscript “Intestinal lipase characterization in common snook (Centropomus undecimalis) juveniles” by Concha-Frías and others presents some aspects of lipase activity in the crude homogenate of the whole intestine of starved fish.  

The rationale behind the study is that an increasing population need more food and therefore it is necessary to culture this species. To achieve this, the auteurs claim that knowledge on the digestive lipase activity is asked for. Why knowledge on temperature optimum for digestive enzymes, in vitro is important for optimal feeding of fish in captivity remains unanswered. Nor does the manuscript offer any arguments to why the effect of lipase blockers on the lipase in question should benefit the production of this species. On the other hand, one may argue that new knowledge is always welcome, even if it has no known application at the present moment.

Overall, the manuscript is not very well written, with numerous sentences that are awkward and difficult comprehend. The manuscript also gives the impression that the authors have a poor overview of lipases and lipases in the teleost fish in particular. By far all lipases are extracellular, as claimed in line 52. Consequently, a homogenate of the whole intestine will present a large array of lipases like such as ATGL, HSL and MGL, that are associated with cytosolic lipid droplets. However, I acknowledge that the major lipase present in such a homogenate will come from the pancreatic tissue associated with the pyloric caeca and that would be BAL also called CEL. Advanced species like this does not have functional pancreatic lipase nor colipase, this is well established by now.

If the object is to find the optimal dietary lipid level. Feed the fish different lipid levels and look at the effect on the organism. Even if the fish may digest and absorb large amount of fat, doesn’t mean this is optimal for the organism. Knowing the optimal temperature for the fish is useful, however, this usually har little correlation with in vitro temperature optimum for enzymes.

If the object was to identify the digestive lipases, the auteurs should work on intestinal content and/or more targeted dissections of pancreatic tissue. Such substrates will also contain multiple lipases, but the endogen digestive lipases will be found in much higher concentrations.   

Reviewer 3 Report

See attached
